# Analysis of Game Actions and Performance in Young Soccer Players: A Study Using Sequential Analysis

Alejandro Sabarit [1], Juan P. Morillo-Baro [1], Rafael E. Reigal [1], Juan A. Vázquez-Diz [2], Antonio Hernández-Mendo [1] and Verónica Morales-Sánchez [1,*]

1    Faculty of Psychology, University of Malaga, 29071 Malaga, Spain
2    Faculty of Sport Sciences, University of Seville, 41640 Osuna, Spain
*    Correspondence: vomorales@uma.es

**Abstract:** The objective of this research is to analyze the performance of actions in a reduced game situation in a sample of young soccer players. This is a game format in which possession of the ball has to be maintained. The sample consisted of 85 young players aged between 12 and 16 years, observing a total of 58 reduced games and using an analysis instrument called the Game Performance Assessment Instrument (GPAI). The essential data quality analyses are carried out, including the use of correlation indexes, Cohen's Kappa and the Phi index for intra- and inter-observer reliability. Generalizability and invariance analyses are also performed to estimate the reliability, validity and precision of the numerical structure and its generalizability to other samples or populations with similar distributions. An observational design of nomothetic, punctual and multidimensional types is used, and subsequently, a sequential analysis of the game actions is carried out from the observations. The results show significant relationships between decision-making behaviors (adequate and inadequate) and technical execution behaviors (adequate and inadequate). The findings have highlighted a clear relationship of interdependence between technical execution and decision making. This information is very useful for the design and planning of training oriented to the optimization of sport performance in soccer.

**Keywords:** decision making; technical execution; sport performance; soccer

## 1. Introduction

Soccer is a team sport that takes place in a common and dynamic context, where reciprocal relationships occur between the behaviors of the participants and the socio-physical environment [1,2]. Despite this scenario being characterized by the appearance of a multitude of interactions, which seems to imply a situation of randomness and uncertainty in the development of the game, several studies indicate that the synchronized interaction of the aspects that constitute it allows for carrying out an orderly sequence of behavioral patterns. This indicates that the game presents a certain sequential interdependence of the player's participation in the game, which would give rise to behaviors with a greater probability of occurrence than those offered by chance [3,4]. So, it is of great importance to know the characteristics of the game, as well as the behavioral patterns of its members, in the context of describing and explaining the individual and collective performance of the team in order to develop scientific and quality knowledge [5–8].

Currently, research that highlights the importance of a rigorous and detailed analysis of the game in soccer, focused on the context, spontaneity of behavior and development of valid and reliable observation instruments to explain the effectiveness of player behavior has been multiplied [9,10]. The observational methodology is ideal for the study of the development of the game in soccer, whose objective is to analyze the actions in their context and habitual dynamics [11]. Thus, through this methodology, it is intended to provide, within a scientific framework, a working tool that allows us to consider and

describe the behaviors of the players, individually and the team collectively, in their natural environment [12,13].

From an observational perspective, the sequential lag analysis [4,14–16] facilitates the obtaining of behavioral sequential patterns that allow us to know whether the transition between such actions is due to a probability greater than that determined by chance [17]. Specifically, with the technique of sequential lag analysis, the option of being able to estimate the relationships that occur between the criterion or conditioned behavior and the other categories included in the taxonomic system, known by the name of mating behaviors, is opened. This analysis based on the binomial test can be used in a complementary and independent way a retrospective or backward temporal perspective and/or a prospective or forward perspective. This allows for identifying and describing the relationships established between the different events that occurred during the game [18]. These relationships will be of activation or inhibition. For these relationships to be considered significant, keeping in mind that the first type of error, alpha, is considered as 5% ($p < 0.05$), they must reach a value of $z \pm 1.96$ [11,15,19–21] according to the proposal of Allison and Liker [22].

A recurrent and useful tool for the observation of behaviors of interest in small-sided game situations in collective sport is the Game Performance Assessment Instrument (GPAI), designed to assess decision making and sports performance [23]. Its aim is to assess the behaviors that occur during the game to demonstrate the technical skill and tactical knowledge of the players. For this purpose, it analyzes individual components (decision making, technical execution and support) and overall game performance (game participation and performance) [24]. The use of reduced games in training is increasingly supported, both under ball and non-ball conditions. Likewise, it is possible to have control of different variables that affect the development of the game and influence the responses of the players, such as the modification of rules, dimensions of the space, number of participants, duration of the task, etc. This makes it possible to manage different fundamental aspects for the development of the physical capacity, technical skills and tactical aspects of soccer players. The evolution of new observation tools allows a deeper study of physical, technical-tactical and psychological characteristics, among others [25].

Regarding decision making in sports, it depends on individual, contextual and task factors [26,27]. Each context is different, and open-ended sports are subject to continuous changes that modify the problem raised. There are several studies that through observational methodology have tried to determine the degree of interaction of a team through the activation-inhibition relationship between lines and with respect to the opponent [8,28] and space management [2,12,29]. On the other hand, decision making involves a process by which people deal with information coming from the environment to develop a motor action [30]. In the study of decision-making processes, although there are different avenues of research from which this process is approached, in this case, the perspective of the deliberate practice of the sport itself stands out, a procedure that, according to Marteniuk [31], is explained by the mechanisms of perception-decision-execution [32].

The interaction between decision making and technical execution can provide valuable information to improve the teaching and learning processes of sports, influencing the optimization of sports performance. However, numerous studies have analyzed these factors in an isolated way or in a complementary way. Few have explored the interaction between them, ignoring an issue that is closely related. If we consider adequate decision making as the result of a complex process that involves different processes, one of the most important issues for its final result is adequate technical execution. Therefore, the relationships between decision making and technical execution need to be evaluated in depth. Thus, the aim of this study was to analyze the performance of game behaviors developed during a situation of ball possession in reduced play in a sample of young soccer players. For this purpose, sequential analysis is used, from which behavioral sequential patterns are generated, mainly focused on decision making and technical execution as criteria behaviors. In this way, it is possible to observe the previous and subsequent behaviors associated with these actions, which make up the taxonomic system. In addition, an analysis of data quality

and generalizability was carried out in order to ensure the reliability, validity and accuracy of the study. Thus, this research studies in more detail the relationship between decision making and the technical execution of in-game actions and highlights the study of sport from an ecological perspective through the observational methodology.

## 2. Materials and Methods

### 2.1. Design

The observational design used in this research is nomothetic, punctual and multidimensional, located in quadrant III (P/I/M) [9]. The observational tool used is a category system instrument, known as GPAI [23].

### 2.2. Participants

The sample consisted of 85 male participants belonging to the infantil (under-14; n = 51) and cadete (under-16; n = 34) categories, aged between 12 and 16 years, belonging to a team from Rincon de la Victoria (Spain). A total of 58 reduced games were observed.

### 2.3. Instruments and Measurements

The HOISAN program (Herramienta de Observación de las Interacciones Sociales en Ambientes Naturales) [33,34] was used for the registration and coding of observations, data quality analysis (correlation indexes, Phi Index and Cohen's Kappa) as well as sequential analysis. The SAGT program (Software Application for Generalizability Theory) [35] was also used for the generalizability analysis, through which the models are determined to ensure the validity, reliability and precision of the intra- and interobserver agreements, the homogeneity of the categories and the minimum estimate of sessions of the sample design that allows accurate generalization. The Microsoft Excel© program was used to estimate the invariance, considering that if Ho: it is assumed that the correlations are equal and, therefore, the invariance is confirmed, and if H1: it is assumed that the correlations are different and, therefore, there is variance. This allows us to assume that the observational tool is optimal, in this case, regardless of the age of the observed players. Finally, regarding the sequential analysis, it is considered that if Ho: it is assumed that there are no behavioral patterns since all categories have the same probability of occurrence, and if H1: the existence of behavioral patterns will be determined with a probability greater than determined by chance for behaviors associated with decision making.

The observation tool used is formed by six criteria (attacking player, support player, decision making, technical execution, support and participating team) and 20 categories (see Table 1).

### 2.4. Procedure

First, observation and data collection were carried out by two different observers. The unit of analysis was the action of the players in ball possession phase in small-sided games, recording the decision making, technical execution and support in each event. After data collection, data quality analysis was performed. For this purpose, Pearson, Spearman and Kendall's Tau-b correlation coefficients were obtained, as well as Cohen's Kappa and Phi concordance indexes. All this process was carried out using the HOISAN program.

A generalizability analysis was then carried out, for which reliability, homogeneity of the categories and sample estimation analyses were performed using the SAGT program.

The next step was to perform an invariance analysis using Microsoft Excel©.

Finally, the sequential analysis of lags was performed, which allows us to obtain sequential patterns of prospective and retrospective behavior, showing a succession of behaviors greater than expected by chance. The lags from −1 to −5 show a retrospective or backward perspective, while 1 to 5 show a prospective or forward perspective (1 or −1 being very short, to 5 or −5 being very long). In this case, the criterion behaviors analyzed were adequate and inadequate decision making, as well as adequate and inadequate technical execution. For this purpose, the infantil and cadete categories were used as a set.

Finally, those transitions that obtained values higher than 1.96 were selected. This analysis was also carried out using the HOISAN program.

**Table 1.** Observation tool used created from the 'GPAI'.

| Criteria | Categories |
| --- | --- |
| Attacking player | Player Attack 1 of team 1 (JA1E1) |
| | Player Attack 2 of team 1 (JA2E1) |
| | Player Attack 3 of team 1 (JA3E1) |
| | Player Attack 1 of team 2 (JA1E2) |
| | Player Attack 2 of team 2 (JA2E2) |
| | Player Attack 3 from team 2 (JA3E2) |
| Support player | Player Support 1 of team 1 (J1E1_Ap) |
| | Player Support 2 of team 1 _(J2E1_Ap) |
| | Player Support 3 of Team 1 (J3E1_Ap) |
| | Player Support 1 of team 2 (J1E2_Ap) |
| | Player Support 2 of team 2 (J2E2_Ap) |
| | Team 2 Support Player 3 (J3E2_Ap) |
| Decision making | Appropriate decision making (ADD) |
| | Inadequate decision making (IDT) |
| Technical execution | Adequate technical execution (ATE) |
| | Inadequate technical execution (ETI) |
| Support | Adequate support (AA) |
| | Inadequate support (IA) |
| Participating team | Team 1 (E1) |
| | Team 2 (E2) |

The study was always conducted respecting the ethical principles established in the Declaration of Helsinki by the World Medical Association [36] and complying with the Standards of Ethics in Sport Science Research [37]. To this end, prior to data collection, informed consent was obtained from fathers and/or mothers, as well as voluntary acceptance from all participants. In addition, the study was approved by the Ethics Committee of the University of Malaga (19-2015-H).

## 3. Results

### 3.1. Data Quality Analysis

The data quality analysis shows the agreement between the different observations made, either by different observers or by the same observer in different sessions and, therefore, the reliability of the results obtained. For this purpose, Pearson, Spearman, Kendall's Tau b correlation coefficients, as well as Cohen's Kappa and Phi concordance indexes are obtained.

The results obtained show high levels of agreement for the first and second observers, respectively, in different categories. Table 2 shows the following values for the first observer: Pearson 0.99, Spearman 0.94–0.97, Kendall's Tau b 0.89–0.95, Cohen's Kappa 0.89–0.95 and Phi coefficient 0.88–0.94 with a permitted range of $\pm 5$ frames in the latter two cases. Likewise, Table 3 shows the following values for the second observer: Pearson 0.99, Spearman 0.93–0.97, Tau b de Kendall 0.89–0.94, Kappa de Cohen 0.86–0.90 and Phi coefficient 0.85–0.90.

**Table 2.** Correlation coefficients and concordance indices for intraobserver reliability as a function of different sensitivity intervals (O1).

| | | | | | Cohen's Kappa | Phi |
|---|---|---|---|---|---|---|
| **Intra O1 Concordance** | | | | | | |
| | **Pearson** | **Spearman** | **Kendall (F)** | **Kendall (I)** | **±5** | **±5** |
| Under 16 1 | 0.99 | 0.97 | 0.93 | 0.93 | 0.90 | 0.90 |
| Under 16 2 | 0.99 | 0.96 | 0.95 | 0.95 | 0.93 | 0.93 |
| Under 14 1 | 0.99 | 0.94 | 0.89 | 0.89 | 0.90 | 0.89 |
| Under 14 2 | 0.99 | 0.94 | 0.90 | 0.90 | 0.89 | 0.88 |
| Under 14 3 | 0.99 | 0.97 | 0.94 | 0.94 | 0.95 | 0.94 |
| Under 14 4 | 0.99 | 0.96 | 0.93 | 0.91 | 0.93 | 0.92 |

Note: Kendall (F) = frequencies; Kendall (I) = intensities.

**Table 3.** Correlation coefficients, Phi coefficients and Cohen's kappa index for intraobserver reliability (O2) as a function of different sensitivity intervals.

| | | | | | Cohen's Kappa | Phi |
|---|---|---|---|---|---|---|
| **Concordance Intra Observer 2** | | | | | | |
| | **Pearson** | **Spearman** | **Kendall (F)** | **Kendall (I)** | **±5** | **±5** |
| Under 16 1 | 0.99 | 0.97 | 0.94 | 0.93 | 0.90 | 0.90 |
| Under 16 2 | 0.99 | 0.95 | 0.91 | 0.90 | 0.88 | 0.88 |
| Under 14 1 | 0.99 | 0.93 | 0.89 | 0.89 | 0.86 | 0.85 |
| Under 14 2 | 0.99 | 0.95 | 0.91 | 0.89 | 0.88 | 0.86 |
| Under 14 3 | 0.99 | 0.95 | 0.90 | 0.90 | 0.90 | 0.89 |
| Under 14 4 | 0.99 | 0.95 | 0.91 | 0.91 | 0.89 | 0.88 |

Note: Kendall (F) = frequencies; Kendall (I) = intensities.

Values close to or higher to 0.80 in the correlation coefficients and concordance indices demonstrate that there is congruence in the observations made by the observers and, therefore, the reliability of the results obtained.

Table 4 shows that the results obtained reflect high levels of interobserver agreement for the different categories, with the following values: Pearson 0.99, Spearman 0.90–0.94, Kendall's Tau b 0.82–0.89, Cohen's Kappa 0.80–0.84 and Phi coefficient 0.79–0.83.

### 3.2. Generalizability Analysis

The Generalizability Theory, originally by Cronbach et al. [38], attributes the concept of the error to multiple influencing factors or facets and, using the procedures of analysis of variance and experimental designs, can analyze the different sources of variation that affect a measure of observational origin [39]. Generalizability analysis, therefore, has as one of its main objectives to measure the variance components that contribute to errors in a measurement in order to reduce them and to be able to estimate the degree of generalizability of a measurement design [40–42], that is, to demonstrate that the different measurements performed produce similar results. In this study, we aim to estimate the reliability, validity and level components of the optimization design. To determine intra- and inter-observer agreement, a two-facet design, category and observers = C/O, was performed, while for

the determination of category homogeneity, a two-facet O/C design was used. Two-facet design categories and matches = C/P were also used for sample estimation.

**Table 4.** Correlation coefficients, Phi coefficients and Cohen's kappa index for interobserver reliability according to different sensitivity intervals.

| | Interobserver Agreement | | | | | |
| --- | --- | --- | --- | --- | --- | --- |
| | **Pearson** | **Spearman** | **Kendall (F)** | **Kendall (I)** | **Cohen's Kappa** | **Phi** |
| | | | | | **±5** | **±5** |
| Under 16 1 | 0.99 | 0.94 | 0.89 | 0.86 | 0.82 | 0.82 |
| Under 16 2 | 0.99 | 0.92 | 0.87 | 0.85 | 0.81 | 0.80 |
| Under 14 1 | 0.99 | 0.90 | 0.85 | 0.82 | 0.80 | 0.79 |
| Under 14 2 | 0.99 | 0.91 | 0.82 | 0.82 | 0.83 | 0.81 |
| Under 14 3 | 0.99 | 0.94 | 0.88 | 0.88 | 0.84 | 0.83 |
| Under 14 4 | 0.99 | 0.90 | 0.84 | 0.84 | 0.84 | 0.83 |

Note: Kendall (F) = frequencies; Kendall (I) = intensities.

To determine intraobserver reliability, a two-facet, category and observer (C/O) measurement design was used. The objective was to determine the degree of agreement between the observations of the same observer in different sessions. The variability is associated with the categories facet, with 98.04% being null for the observers' facet and 1.96% for the C/O interaction facet. This indicates that the data obtained depend on the categories designed independently of the participating observers since the data do not vary according to them. The relative and absolute generalizability coefficients (G) are 0.99, which determines a high reliability and precision in the generalizability of the results.

Regarding interobserver reliability, the results obtained are very similar to the previous ones. First, a two-facet measurement design, category and observers (C/O), was used. In this case, the objective is to determine the degree of agreement between two observers. The variability is associated with the categories facet, with 98.67% being null for the observers' facet and 1.33% for the C/O interaction facet. As previously mentioned, this indicates that the data obtained depend on the categories designed independently of the participating observers. Finally, the relative and absolute generalizability coefficients (G) are 0.99, indicating a high reliability and precision in the generalizability of the results.

Regarding the validity analysis, a two-faceted observer-category (O/C) design was used. The objective is to determine the homogeneity of the categories; that is, to check the degree to which the categories that make up the taxonomic system differentiate between the different actions. The relative and absolute G coefficients obtained are zero (.00), indicating that the categories have a high degree of differentiation.

Finally, an optimization study was carried out using the following two-facet design: categories and games (C/P). The objective of this analysis is to estimate the minimum number of games necessary to observe in order to be able to generalize the results accurately. The variability is associated with the categories facet with 86.99%, with 4.09% corresponding to the matches facet and 8.92% for the C/P interaction.

Table 5 shows the results of the relative and absolute generalizability coefficients obtained in the optimization study, being 0.99 from 15 games or matches observed, at which point the results can be accurately generalized.

**Table 5.** Generalizability analysis. Design optimization study (C/P) for estimation of the minimum number of matches.

| | | | | | Summary | | | | | |
|---|---|---|---|---|---|---|---|---|---|---|
| **Name of Securities** | **1** | | | | **5** | | | | | |
| Matches | (1; INF) | (5; INF) | (10; INF) | (15; INF) | (20; INF) | (25; INF) | (30; INF) | (35; INF) | (40; INF) | (45; INF) |
| Categories | (18; INF) | (18; INF) | (18; INF) | (18; INF) | (18; INF) | (18; INF) | (18; INF) | (18; INF) | (18; INF) | (18; INF) |
| Total observations | | | | | | 450 | | | | 810 |
| Relative G coefficient | 0.907 | 0.980 | 0.990 | 0.993 | 0.995 | 0.996 | 0.997 | 0.997 | 0.997 | 0.998 |
| Absolute G coefficient | 0.870 | 0.971 | 0.985 | 0.990 | 0.993 | 0.994 | 0.995 | 0.996 | 0.996 | 0.997 |

Figure 1 shows a design optimization study graph (C/P) representing the values of the absolute G coefficient for the facet 'Matches', as a function of the minimum number of matches to be observed. As can be seen, there is a ceiling effect from the third match onwards.

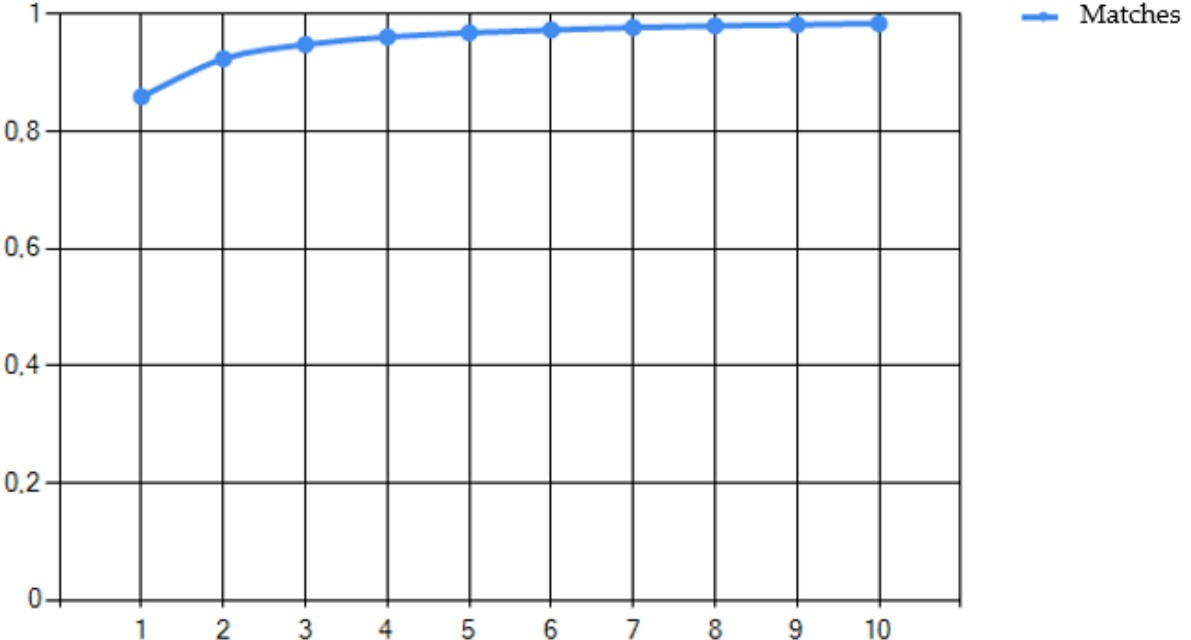

**Figure 1.** Ceiling effect in the design optimization study graph (C/P) representing the absolute G coefficient values for the 'Matches' facet.

### 3.3. Invariance Analysis

Invariance allows us to know whether the meaning of a construct is independent with respect to a group or test, such that the measurement of a construct on different occasions and before different groups provides identical results [43,44]. Therefore, the scores obtained through this analysis are invariant, regardless of the group or the form of the test, since they are interpreted with the same meaning [45].

Table 6 shows the results obtained from the invariance analysis. Since the z-values do not reach values of $\pm 1.96$, we cannot reject the null hypothesis and state that there are differences between groups. Therefore, since the correlations do not differ significantly ($p \geq 0.05$), it is concluded that there is no difference between groups (by observers, in this case) in the use of the observational tool.

**Table 6.** Hypothesis test to evaluate the difference between correlation coefficients in independent samples.

| | Obs. 1 Intra | Obs. 2 Intra | | | | | |
|---|---|---|---|---|---|---|---|
| | r1 | r2 | n1 | n2 | Numerator | Denominator | z |
| Pearson | 0.99 | 0.99 | 58 | 58 | 0 | 0.20 | 0 |
| Tau b Kendall | 0.92 | 0.91 | 58 | 58 | 0.06 | 0.20 | 0.32 |
| Spearman | 0.96 | 0.95 | 58 | 58 | 0.11 | 0.20 | 0.60 |
| Kappa Cohen | 0.92 | 0.90 | 58 | 58 | 0.12 | 0.20 | 0.61 |

*3.4. Sequential Analysis*

The results obtained from the sequential analysis for the actions selected as criterion behaviors are presented below.

Tables 7 and 8 show the sequential patterns of behaviors from a retrospective and prospective perspective for each criterion behavior. In these, one can observe those mating behaviors that are significantly related to each selected criterion behavior. For this, a value equal to or greater than 1.96 must be obtained.

**Table 7.** Sequential analysis for decision making and technical execution criteria behaviors.

| | | | | | TDA | | | | | |
|---|---|---|---|---|---|---|---|---|---|---|
| Code | Ret − 5 | Ret − 4 | Ret − 3 | Ret − 2 | Ret − 1 | Ret + 1 | Ret + 2 | Ret + 3 | Ret + 4 | Ret + 5 |
| JA2E1 | 1.141 | −1.454 | −0.67 | −0.426 | −1.134 | 0.74 | 2.138 | −0.436 | 1.444 | 1.364 |
| ETA | 0.28 | 2.177 | −0.098 | −1.047 | −0.41 | −1.362 | −0.73 | −1.367 | −0.081 | 0.28 |
| | | | | | **TDI** | | | | | |
| Code | Ret − 5 | Ret − 4 | Ret − 3 | Ret − 2 | Ret − 1 | Ret + 1 | Ret + 2 | Ret + 3 | Ret + 4 | Ret + 5 |
| ETI | 0.28 | 2.177 | −0.098 | −1.047 | −0.41 | −1.362 | −0.73 | −1.367 | −0.081 | 0.28 |
| | | | | | **ETA** | | | | | |
| Code | Ret − 5 | Ret − 4 | Ret − 3 | Ret − 2 | Ret − 1 | Ret + 1 | Ret + 2 | Ret + 3 | Ret + 4 | Ret + 5 |
| JA3E1 | −0.466 | 0.814 | −1.552 | 1.976 | −0.282 | 0.522 | −1.399 | −0.611 | 0.039 | 1.443 |
| TDA | 0.28 | −0.081 | −1.367 | −0.73 | −1.362 | −0.41 | −1.047 | −0.098 | 2.177 | 0.28 |
| AA | −0.384 | −0.727 | −0.134 | −0.131 | 0.176 | −0.738 | −0.436 | −0.134 | 1.995 | 0.49 |
| | | | | | **ETI** | | | | | |
| Code | Ret − 5 | Ret − 4 | Ret − 3 | Ret − 2 | Ret − 1 | Ret + 1 | Ret + 2 | Ret + 3 | Ret + 4 | Ret + 5 |
| TDI | 0.28 | −0.081 | −1.367 | −0.73 | −1.362 | −0.41 | −1.047 | −0.098 | 2.177 | 0.28 |
| AI | −0.384 | −0.727 | −0.134 | −0.131 | 0.176 | −0.738 | −0.436 | −0.134 | 1.995 | 0.49 |

Note. JA2E1 = attacking Player 2 of team 1; JA3E1 = attacking Player 3 of team 1; TDA = adequate decision making; TDI = inadequate decision making; ETA = adequate technical execution; ETI = inadequate technical execution; AA = adequate support; AI = inadequate support; Ret= delay.

The results show that the criterion behavior TDA (adequate decision making) is significantly related to the mating behavior JA2E1 (attacking player 2 of team 1) from a prospective perspective and to ETA (adequate technical execution) from a retrospective perspective. In turn, the criterion behavior TDI is significantly related to the mating behavior ETI (inadequate technical execution) from a retrospective perspective.

Regarding the sequential patterns obtained when technical execution is the criterion behavior, it is shown that for the criterion behavior ETA, a significant relationship is established in retrospective perspective with the mating behavior JA2E1, and in prospective perspective with the behaviors TDA and AA (adequate support). On the other hand, the criterion behavior ETI is significantly related to the mating behaviors TDI (inadequate decision making) and AI (inadequate support), both from a prospective perspective.

**Table 8.** Retrospective and prospective sequential patterns of behavior from the criterion behaviors decision making and technical execution, both adequate and inadequate, where Ret is the delay.

| | Sequential Analysis | | | |
|---|---|---|---|---|
| **Category** | **Ret − 4** | **CC** | **Ret + 2** | **Ret + 4** |
| Set | ETA 2.177 | TDA | JA2E1 2.138 | |
| | ETI 2.177 | TDI | | |
| | | ETA | | TDA 2.177 |
| | | | | AA 1.995 |
| | | ETI | | TDI 2.177 |
| | | | | AI 1.995 |

Note. JA2E1 = player attack 2 of team 1; TDA = adequate decision making; TDI = inadequate decision making; ETA = adequate technical execution; ETI = inadequate technical execution; AA = adequate support; AI = inadequate support; Ret = delay.

### 3.5. Decision Making

In relation to decision making, the actions or criteria of adequate decision making and inadequate decision making were considered. These found long sequential patterns (four delays). Specifically, the criterion behavior TDA obtained a long sequential pattern in retrospect with respect to the paired behavior ETA. Consistent with the above, the criterion behavior TDI obtained a long sequential pattern in retrospect with respect to the pairing behavior ETI.

### 3.6. Technical Execution

Regarding the behavioral criterion of technical execution, in the same line as in the decision-making criterion, the actions of adequate and inadequate technical execution were considered. As in the previous case, long sequential patterns were obtained (four delays), but, in this case, from a prospective perspective. Specifically, the criterion behavior ETA obtained a long sequential behavioral pattern in prospective with respect to the mating behavior TDA and AA; while the criterion behavior ETI, obtained a long sequential pattern in prospective in relation to the mating behavior TDI and AI. These results reinforce those found in the previous section.

## 4. Discussion

The main objective of this research was to analyze the performance of actions during small-sided game situations in ball possession tasks in a sample of young soccer players. For this purpose, by means of observational methodology, behavioral sequential patterns of such actions executed during the performances in the small-sided games were analyzed.

First, and starting from the decision-making action as the criterion behavior, the findings show that adequate technical execution precedes and activates the adequate decision-making action. The same occurs with inadequate technical execution and inadequate decision making, where a behavioral sequential pattern occurs in which the former precedes and activates the appearance of the latter. In line with the above, the results have shown the same association based on technical execution as a criterion behavior. In this sense, adequate technical execution activates, from a prospective perspective, the appearance of adequate decision making, in addition to the appearance of adequate support behavior. On the other hand, inadequate technical execution activates the action

of inadequate decision making from a prospective perspective, doing the same with the behavior of mating inadequate support.

These findings are relevant in that decision making is fundamental for performance in soccer. In these contexts, it is necessary to choose the most appropriate game option at the exact moment and be able to execute it [46,47]. In addition, following Araújo [48] and Araújo et al. [49], the study of decision making should start from the analysis of the different characteristics of the sport modality in question, i.e., the context of the game, which is collected in this work. Furthermore, this is defended by González-Villora et al. [30], who state that the success of game actions depends on elements such as the execution of the athlete and other individuals contextual and task factors. In this work, it is observed how technical execution is fundamental for correct decision making. This issue is probably conditioned, as pointed out by the authors described above, by the characteristics of the reduced game, which requires great precision in technical execution for the game actions to be effective. Each decision context is different, and open sports are subject to continuous changes that modify the problem. Even more when working in this type of situation that increases the speed of the game, limits the spaces, increases the incidence of error and conditions the need for continuous support.

The results obtained reinforce the thesis of previous studies showing the existence of significant relationships between decision making and technical execution [50–53]. González-Villora et al. [30,52] found significant correlations between decision making and player actions in young soccer players, based on technical components and motor executions. These studies, moreover, point to the importance of cognitive processes associated with technical and tactical aspects that, although not explicitly analyzed in this research, represent a crucial area of study for the analysis of sports performance and, specifically, decision making. These findings are in line with the model proposed by Marteniuk [31] on motor activity, which is composed of the following three phases: perception, decision and execution. The cognitive factor in decision making and technical-tactical procedures of the game are immersed in this scheme [54,55].

An important issue to highlight is that, although chronologically it can be assumed that the sequential pattern of behaviors would begin with decision making and would be followed by technical execution, some phases of the process are hidden under the limitations of their own nature and the methodology used. Because of this, decision making could be observed from the footprints or behavioral traces of athletes through successful versus unsuccessful patterns [26]. For this reason, decision making and technical execution or motor action would be paired, resulting in a more important degree of association than the chronological order in the sequential pattern of behavior.

Secondly, these findings suggest that both factors (technical execution and decision making) are interdependent. This relationship involves cognitive processes associated with technical-tactical aspects of the sport in question [56]. So, the influence and implications of one on the other should be considered in the design and planning of exercises oriented to their training and development. A good decision can be useless if the technical execution does not accompany it, and vice versa, good technical actions can be unsuccessful if the decision is not correct or executed at the right time. Therefore, in the face of sports training at the technical and decisional level, involving the decisional aspect in isolation from the context is inappropriate, so the integration of activities that associate cognitive components with specific actions of the game, both technical and tactical, is proposed [46]. Instead of working in a partial and isolated way, representative tasks of the game should be proposed for a better coupling between perception and action [57]. In recent years, sports education is moving towards a model that combines or integrates the cognitive, decisional and motor components in which the person perceives, decides and executes motor skills [55], so interventions are being designed in decision making based on cognitive aspects and linked to technical actions to promote a better response to game situations [58,59]. For its correct evaluation and intervention, it should be based on an ecological model based on the components of the task, individual and environment [49], since decision making is

conditioned by the context of the game [51,53]. That is, its study must start, in addition to the individual characteristics of the athlete, from the different characteristics of the sport modality, the task and the environment [30,48,60] considering varied contexts [58,59].

## 5. Conclusions

The present study has analyzed the relationship between decision making and technical execution as evaluated through actions in small-sided games aimed at ball possession. For this purpose, an observational methodology has been used; specifically, a sequential analysis of such behaviors has been carried out. The results obtained through this analysis show a relationship of interdependence between both behaviors, which points out the importance of the cognitive processes associated with technical-tactical aspects. For this reason, it is evolving into a training model that integrates the cognitive component for the development and optimization of sports performance.

However, we must keep in mind that this study has been carried out exclusively on a sample of young males. An investigation that includes a group of girls would allow us to identify whether there are differences between genders. In addition, we should keep in mind that, in a sport with an open and dynamic context such as soccer, it would be necessary to analyze such relationships in different situations, modifying the environment.

In this sense, with a view to future lines of research, it would be interesting to study the relationships between decision making and technical execution using small-sided games that involve different manipulations of the environment that are adjusted to the demands of competition. Although this tool is suggested to be valid and suitable for decisional and technical-tactical evaluation [61,62], different authors have pointed out the importance of including goal orientation in small-sided games (games modified by representation) as opposed to the ball conservation model (games modified by exaggeration), since it allows to experience a greater number of real and specific game situations, as well as to justify the effectiveness of reduced games on the athlete's learning [55]. Likewise, it would be interesting to use complementary analyses, such as polar coordinates, to deepen the findings obtained. Finally, it could also be of great interest to analyze the influence of other cognitive and psychological variables in the decision-making process and technical-tactical actions, such as stress, processing speed or reaction time, among others.

**Author Contributions:** A.S., R.E.R., J.P.M.-B., V.M.-S. and A.H.-M. participated in the study design and data collection, performed statistical analyses and contributed to the interpretation of the results, wrote the manuscript, and approved the final manuscript as presented. All authors made substantial contributions to the final manuscript. All authors have read and agreed to the published version of the manuscript.

**Funding:** This research received no external funding.

**Institutional Review Board Statement:** The study was conducted in accordance with the Declaration of Helsinki and approved by the Ethics Committee of the University of Malaga (19-2015-H).

**Informed Consent Statement:** Informed consent was obtained from all subjects involved in the study. Written informed consent has been obtained from the patient(s) to publish this paper.

**Data Availability Statement:** Not applicable.

**Conflicts of Interest:** The authors declare no conflict of interest.

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
