# Peer review of "Analysis of Game Actions and Performance in Young Soccer Players: A Study Using Sequential Analysis"

_sustainability, doi:10.3390/su142013263_

Round 1

Reviewer 1 Report

The present study is carried out with the aim of finding empirical evidence of the close relationship between the success in decision-making and the quality of the execution of certain technical actions typical of a sport, soccer. The design of the research and the methodology used are outstanding elements of the study. They are described very carefullly and the most modern and advanced models, methods and computer programs are used, which allow the authors to offer multiple proofs of the reliability and validity of the results achieved.

Accepting that the objectives of the study, the methodolgy used, the statistical analyzes and the results achieved are appropriate and reliable, our comments are rather aimed at helping the authors improve certain less relevant formal aspects, which are indicated below.

1. Check a space and a comma in line 121 “the program was used to estimate the invariance, considering that if Ho : , it is assumed that ..”.

2. Check for a space on line 122 between H1 and the colon “correlations are equal and therefore invariance is confirmed, and if H1 : is…”.

3. Review the word “Partidos” to the right of Figure 1.

4. As the study is read, it is intuited that there are several observers, but it is not known how many there are until the results section is reached. Perhaps the number of reviewers should be specified earlier, somewhere dedicated to the description of the research methodology.

5. Carefully review the References so that they all have the same format and structure. The last author must be preceded by a comma. Sometimes it does that, and sometimes it doesn't. For example, in References 1, 2, and 3 the comma is put, but in References 4 and 7, it is not. Review all References. When you reach the last author, you must put the "&" symbol. In Reference 14 it is not put. And in Reference 19 the symbol "y" is put. Review.

Author Response

Thank you for your feedback. We have replied to your suggestion.

Reviewer 2 Report

Dear respected authors,

1.      The study's main aim and content have been reflected well in the Abstract section.

2.      Based on the Keywords list, it is suggested to unify “decision making” as it is, or as “decision-making” in whole the text. In the case of using the latter one, it should be reflected in the Keywords list.

3.      The aim of the study, the necessity of having (publishing) this study, and the research gap based on the related literature should be highlighted in the Introduction section.

4.      The definition of “soccer” in the first sentence of the Introduction section should be revised. If I am not mistaken, “soccer” is a game, not a team. The respected author probably aimed to define the “Soccer team”. It is suggested to recheck this sentence.

5.      The cohesion in the Introduction sections, especially for connecting each sentence to its next one, should be improved.

6.      Forgetting about the support of the editorial office before each publication, it is expected from the respected authors to use the correct format for the referencing in the text. For instance, in Line 42 it is better to write [5-8], instead of [5,6,7,8]; or using [4, 14-16] instead of [4,14,15,16] in Line 52. In addition, it is expected to use the references in the text, in order, according to the list of references at the end of the manuscript. Please check the whole text according to this hint.

7.      Defining Acronyms/Abbreviations and using them, is appropriate for those phrases that have been mentioned frequently in the text. According to this issue, defining “OM” for the “observational methodology” in Line 47 is meaningless, especially when it has not been used in any other part of the article instead of the complete phrase; similarly, defining “GT” for “Generalizability Theory “ in Line 177. The whole manuscript should be rechecked for the other Acronyms/Abbreviations considering this issue.

8.      In Line 63, it is suggested to mention that the first type of error, alpha, is considered as 5%, therefore, the test statistic value should be in the interval of Z plus/minus 1.96.

9.      To perform scientific research, it is highly recommended to use scientific statistical software like SPSS, Minitab, AMOS, etc. to evaluate such hypothesis tests as mentioned in this manuscript. All the hypothesis tests should be reflected in the Materials and Methods section separately. Using such applications empowers the respected researchers to find a relation not in 95% significant intervals, but also in 99%, and even more. The obtained results by them give deeper information about the performed analysis.

10.  In sub-section 3.1, and generally, first the obtained results should be presented. Then by getting a conclusion from the results, the agreement/disagreement about an issue should be mentioned. Additionally, the content of Tables 2 to 4 should be explained in detail in this section.

11.  The number of the tables/figures while there is an explanation for them in the text should be rechecked. For instance, Table 4, should be modified to Table 5 in line 215.

12.  The hypothesis of the Invariance Analysis and its related null and alternative hypothesis should be mentioned and explained in the Materials and Methods section, and only the obtained results should be reflected in the Result section. Therefore, sub-section 3.3 needs to be modified.

13.  The last column in Table 6, column Z, needs more explanations in the text.

14.  Table 8 should be retyped in a better way. The unnecessary columns can be deleted from his table.

15.  The Conclusion section should briefly contain the aim(s), used tools and methods, and should have a separate paragraph for the limitations of the study as well as the potential future studies. It is recommended to extend this section considering these issues.

16.  The text needs a moderate grammatical check. 

Author Response

Thank you for your feedback. We have replied to your advice.

Round 2

Reviewer 2 Report

Dear respected authors,

The methodology, procedures, and steps of the research have been done logically. I appreciate your efforts and patience to answer the comments and modify the manuscript accordingly. The manuscript's revised version is worth getting published in the respected journal.